

# A novel evaluation approach for functional impairment in subacromial impingement syndrome: focus on Temporal Summation of Activity-related Pain (TSAP)

Katsuyoshi Tanaka[1], Shota Oda[2] and Masashi Izumi[2,3]

[1] Department of Physical Therapy, Bukkyo University, Kyoto, Kyoto, Japan
[2] Department of Rehabilitation, Kochi Medical School Hospital, Nankoku, Kochi, Japan
[3] Department of Orthopaedic Surgery, Kochi Medical School, Kochi University, Nankoku, Kochi, Japan

## ABSTRACT

**Background**. Patients with subacromial impingement syndrome (SIS) often experience increased pain during repetitive upper-extremity movements in their daily life. However, conventional assessments of movement-evoked pain are mostly limited to single movement pain (SMP) and may not fully capture the effects of pain during repetitive activities. In this study, we developed the Temporal Summation of Activity-related Pain (TSAP) as a novel method for assessing increases in pain intensity during repetitive movements, and investigated its clinical usefulness.

**Methods**. Thirty patients with SIS were included in this cross-sectional study. Movement-evoked pain was assessed using patient-reported outcome measures, SMP and TSAP scores, which were evaluated by the increase in pain intensity after 10 repetitions of a shoulder abduction task. Additionally, the conventional temporal summation of pain (cTSP) was assessed using pinprick stimulation. We analyzed the association between the parameters and the impact of movement-evoked pain on upper extremity dysfunction assessed using Quick Disabilities of the Arm, Shoulder, and Hand (QuickDASH).

**Results**. The TSAP score significantly correlated with the cTSP and QuickDASH scores ($p < 0.05$). Regression analyses revealed that the TSAP score was the only significant factor explaining the impact on upper extremity dysfunction, even after controlling for confounding factors [$B$ (95% CI) = 0.461 (0.099–0.824), $p = 0.015$].

**Conclusions**. TSAP provides valuable insights into the functional impact of pain in patients with SIS. Our findings suggest that TSAP may offer a more sensitive evaluation of movement-evoked pain compared to conventional assessments, although further validation is needed.

Corresponding author
Katsuyoshi Tanaka,
katsutanaka7@gmail.com

## INTRODUCTION

Movement-evoked pain can be characterized as the sensation of pain that occurs during physical activity within a particular context (*Graven-Nielsen & Arendt-Nielsen, 2010*; *Fullwood et al., 2021*). This is a significant concern in patients with chronic shoulder pain, especially those with subacromial impingement syndrome (SIS). Individuals with SIS often experience increased pain during daily activities, which exacerbates their condition and limits functional capacity (*Lewis et al., 2015*). This typical movement-evoked pain is largely attributed to the mechanical stress of the rotator cuff tendons and the surrounding bursa between the acromion and humeral head, leading to inflammation and further discomfort (*Littlewood et al., 2013*). Although this is a significant barrier to effective rehabilitation, conventional assessments of movement-induced pain are mostly limited to single-movement pain (SMP), which may not adequately capture the complexity of pain experienced during functional activities (*Diercks et al., 2014*).

The conventional temporal summation of pain (cTSP), which refers to the experimental temporal summation of sensory stimuli, describes the phenomenon in which the intensity of pain progressively increases in response to repeated exposure to noxious stimuli (*Overstreet et al., 2021*; *Deegan et al., 2024*). Previous studies have shown that individuals who exhibit hyperalgesic responses to repeated instances of evoked pain may be at an increased risk of negative pain-related outcomes (*George et al., 2006*; *Weissman-Fogel et al., 2009*). However, there is a limitation in interpreting the temporal summation during movement because cTSP is conducted using experimental sensory stimuli. It is essential to consider the temporal summation of the sensory stimuli that occur specifically during movement. Considering the significance of evaluating pain during functional and repetitive movements, a comparable methodology specifically designed for the upper limbs may yield important insights into shoulder disorders such as SIS.

Given these limitations, there is a clear need for a novel evaluation method that captures both repetitive and functional aspects of movement-evoked pain in clinical contexts. TSAP addresses this gap by incorporating temporal summation into activity-based pain evaluation, which may help clinicians better understand the relationship between pain and function in shoulder disorders.

Therefore, in this study, we aimed to develop a novel assessment method, termed the Temporal Summation of Activity-related Pain (TSAP), to evaluate movement-evoked pain using repetitive upper-limb tasks in patients with SIS. This method reflects the temporal sum of pain during movement. By capturing both the physical and perceptual aspects of pain during functional shoulder movements, this approach may provide a novel indicator for assessing shoulder function, possibly leading to more effective rehabilitation strategies.

We hypothesized that TSAP would reflect activity-related pain more sensitively than conventional movement-evoked pain assessment, and that it would be associated with dysfunction in individuals with SIS.

## METHODS

### Participants

This study was conducted in accordance with the principles of the Declaration of Helsinki. Sample size was calculated using G*power (v.3.1), which indicated that a total sample size of 26 people would be needed (effect size is 0.5 with 80% power and alpha at .05). After obtaining written informed consent from the participants prior to the study, 30 individuals with chronic SIS aged 48–80 years participated in this cross-sectional study. SIS was diagnosed by orthopedic physicians. The inclusion criteria were as follows: (1) persistent pain for more than 3 months and (2) a positive result on at least one of the orthopedic tests performed by a physical therapist, including the Hawkins impingement sign, Neer's impingement sign, painful arc sign, and/or Empty Can Test (*Michener et al., 2009*). Although previous study (*Michener et al., 2009*) has suggested that the diagnostic accuracy for subacromial impingement syndrome increases when multiple tests are positive, in this study, a positive result on at least one orthopedic test was considered sufficient when supported by clinical judgment to better reflect real-world clinical practice. Exclusion criteria were individuals diagnosed with cancer, brain or spinal cord injury, neurological diseases, or dementia were excluded from the study. In addition, individuals who could not abduct their shoulders by more than 90° were excluded. Furthermore, individuals with other upper-extremity conditions that could affect pain or function were excluded from the clinical assessment. Ethical approval was obtained from the Human Research Ethics Review Committee of the Bukkyo University (2023-18-A).

### Procedures

Demographic data (age, sex, height, weight, duration of pain, and clinical diagnosis) were collected from medical records of the participants. The mean intensity of pain during daily activities in the previous week was assessed using a visual analog scale (VAS) that ranged from 0 (no pain) to 100 (worst pain imaginable). Experienced physical therapists evaluated SMP, TSAP, cTSP, range of motion (ROM) and muscle strength of the affected shoulder. The Central Sensitization Inventory-9 (CSI-9) and Quick Disabilities of the Arm, Shoulder, and Hand (QuickDASH) were obtained from self-reported questionnaires. The details of each assessment are as follows.

#### *Single-movement-pain and TSAP*

The TSAP concept was developed based on previous studies (*Sullivan et al., 2009*). The TSAP procedure is illustrated in Fig. 1. We defined the movement of 90° shoulder abduction in the scapular plane as the task movement. First, the minimum load that induced pain during task movement was identified using weights ranging from 0 to 3 kg. The participants performed 10 repetitions of the task using the identified load, maintaining a pace of one repetition per second. The pain intensities at the first and tenth repetitions were recorded using a 100-mm VAS, and the first repetition was used as the SMP. The difference between the first and tenth pain intensities was calculated using the TSAP score (tenth pain intensity minus the first pain intensity). SMP has been used in previous studies as a measure of movement-evoked pain (*Wan et al., 2018*; *Wang et al., 2023*).

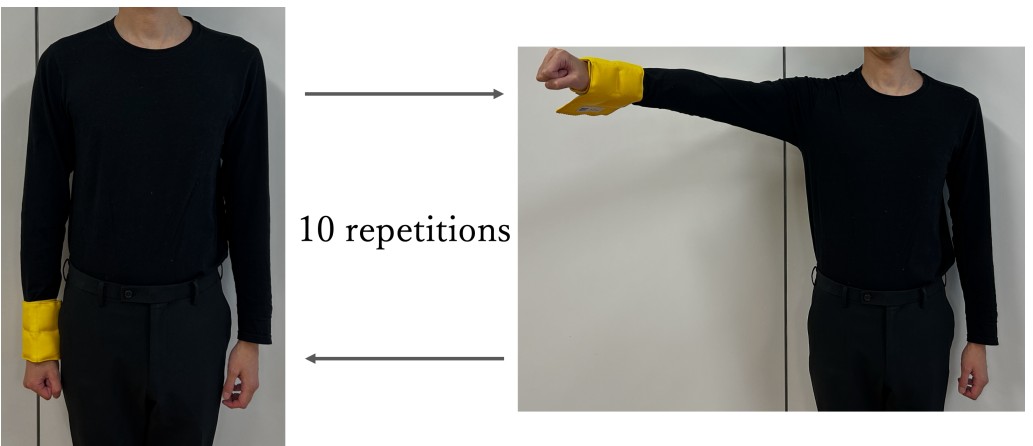

**Figure 1  Procedure of TSAP.** The pain intensity of the first was SMP, and the difference between the first and tenth one was the TSAP score.

### Conventional temporal summation of pain

cTSP was measured using "pinprick" of QuantiPain®, which demonstrated acceptable test-retest reliability and assessment validity with the sensitivity to separate patients with painful knee osteoarthritis from healthy controls (*Izumi et al., 2022*). Repeated pinprick stimuli (60 g) were applied to the bilateral deltoids, and the participants rated the pain intensity of the first and tenth stimuli using a VAS. The cTSP was assessed by calculating the difference in pain intensity between the first and tenth stimuli (Fig. 2).

### Range of motion and muscle strength

Shoulder abduction ROM in the scapular plane was measured. Isometric maximal voluntary strength at 90° in the scapular plane was measured using hand-held dynamometer ($\mu$-Tas F1; ANIMA Inc, Tokyo, Japan). Three tests were performed, and the mean values were used for the analyses.

### Central Sensitization Inventory-9

The CSI has been translated into Japanese and validated (*Tanaka et al., 2017*). CSI is commonly used to measure central sensitization-related symptoms including somatic and emotional symptoms. CSI-9 was developed as a short version of CSI (*Nishigami et al., 2018*). It consists of nine CSI items, each scored from 0 to 4, with a higher total score reflecting higher CS symptomatology. The first author (K.T.) developed the CSI and CSI-9 as part of a previous project and hold the copyright to use it in this study (*Tanaka et al., 2017*; *Nishigami et al., 2018*).

### QuickDASH (disability of the arm, shoulder, and hand)

QuickDASH is a widely used self-report questionnaire for assessing upper extremity function in individuals with shoulder pain (*Vaquerizo et al., 2023*). It comprises 11 items that span six domains (daily activities, symptoms, social function, work function, sleep, and confidence) (*Roe et al., 2013*). Each item was evaluated using a 5-point Likert scale,

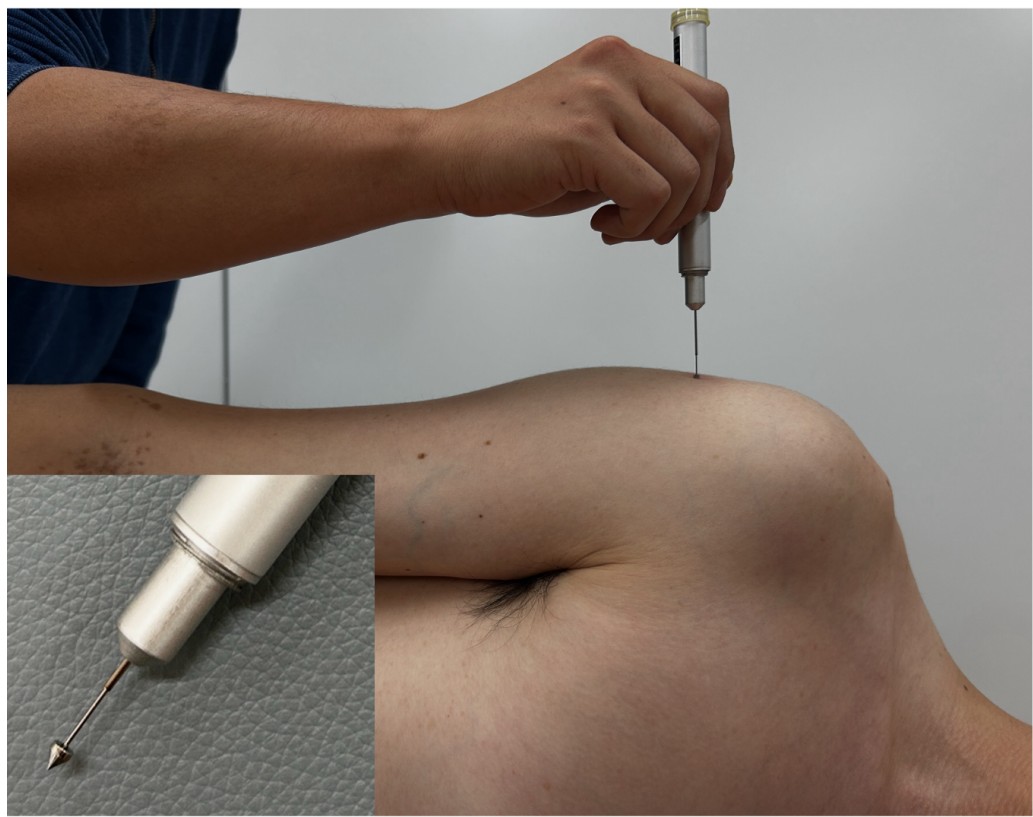

**Figure 2 Assessment of the cTSP.** The difference of pain intensity between first and tenth at bilateral deltoid muscle was cTSP.

with scores ranging from 1 (no difficulty) to 5 (unable). The score was converted to a 100-point scale, where 100 represented the greatest degree of disability. The validated Japanese version of the QuickDASH was used in this study (*Imaeda et al., 2006*), which is freely available for use.

## Statistical analyses

Correlations between the TSAP and pain VAS scores during daily activities and the SMP, cTSP, CSI-9, and QuickDASH scores were analyzed using Spearman's correlation coefficients. In addition, we conducted linear regression analyses using QuickDASH as the dependent variable and pain VAS scores during daily activities, SMP, and TSAP scores as the independent variables (crude model). We completed an adjusted model controlling for demographic and individual functions (ROM of shoulder abduction and isometric muscle power of shoulder flexion). All statistical analyses were performed using SPSS version 26.0 (IBM Corp., Armonk, NY, USA). A significance level of $p < 0.05$ was used for the statistical analyses.

**Table 1   Characteristics of the participants.**

|  | Mean ± SD or N (%) |
|---|---|
| Age (years) | 66.0 ± 10.4 |
| Female [n (%)] | 14 (46.7%) |
| Height (m) | 1.60 ± 0.11 |
| Weight (kg) | 61.9 ± 11.0 |
| Duration of pain (month) | 17.9 ± 25.6 |
| Diagnoses |  |
| Rotator cuff tear | 21 (70.0%) |
| Cuff tear arthropathy | 4 (13.3%) |
| Periarthritis of the shoulder | 3 (10.0%) |
| Other conditions | 1 (6.7%) |
| Pain VAS during daily activities (mm) | 57.8 ± 23.3 |
| SMP (mm) | 23.1 ± 16.7 |
| TSAP score (mm) | 21.1 ± 19.4 |
| cTSP at deltoid muscle |  |
| affected side (mm) | 25.8 ± 28.4 |
| control side (mm) | 21.1 ± 22.0 |
| ROM of shoulder abduction (°) | 124.0 ± 31.9 |
| Muscle strength of shoulder abduction (N) | 46.3 ± 37.9 |
| CSI-9 (point) | 13.8 ± 6.3 |
| QuickDASH (point) | 31.2 ± 19.1 |

**Notes.**

Values are numbers (percent values) for categorical variables and mean ± SD.

Abbreviations: VAS, Visual analogue scale; SMP, single movement pain; TSAP, temporal summation of activity-related pain; cTSP, conventional temporal summation of pain; ROM, range of motion; CSI-9, Central Sensitization Inventory-9; QuickDASH, Quick disability of the arm, shoulder, and hand.

## RESULTS

### Characteristics of the participants

A summary of the participants' demographic and clinical characteristics is presented in Table 1. The mean scores (± standard deviation (SD)) for the SMP and TSAP score were $23.1 \pm 16.7$ and $21.1 \pm 19.4$ mm, respectively.

### Correlations between pain parameters and questionnaires

As shown in Table 2, the TSAP score was significantly correlated with bilateral cTSP (affected side: $\rho = 0.70$, $p < 0.001$; non-affected side: $\rho = 0.66$, $p < 0.001$) and QuickDASH ($\rho = 0.44$, $p = 0.016$), but not with CSI-9. In addition, the QuickDASH score were significantly correlated with bilateral cTSP (affected side: $\rho = 0.47$, $p = 0.01$; non-affected side: $\rho = 0.40$, $p < 0.03$) and CSI-9 ($\rho = 0.41$, $p = 0.03$).

### Comparison of the impacts of movement-evoked pain parameters on QuickDASH

The impact of movement-evoked pain parameters on the QuickDASH score is shown in Table 3. In the crude model, both the SMP ($B$ (95% CI) $= 0.368$ ($0.013-0.724$), $p = 0.043$)

**Table 2 Correlations between pain parameters and questionnaires.**

|  | 1 | 2 | 3 | 4 | 5 | 6 |
|---|---|---|---|---|---|---|
| 1. Pain VAS during daily activities | – | | | | | |
| 2. SMP | .29 | – | | | | |
| 3. TSAP score | .29 | .04 | – | | | |
| 4. cTSP (affected side) | .41[*] | −.01 | .70[**] | – | | |
| 5. cTSP (non-affected side) | .38[*] | .11 | .66[**] | .80[**] | – | |
| 6. CSI-9 | <.01 | .33 | .07 | .22 | .21 | – |
| 7. QuickDASH | .35 | .34 | .44[*] | .47[**] | .40[*] | .41[*] |

**Notes.**

All correlations are Spearman's $\rho$ values. All correlations are two-tailed.

Abbreviations: VAS, Visual analogue scale; SMP, single movement pain; TSAP, temporal summation of activity-related pain; cTSP, conventional temporal summation of pain; CSI-9, Central Sensitization Inventory-9; QuickDASH, Quick disability of the arm, shoulder, and hand.

[*]$p < .05$.

[**]$p < .01$.

**Table 3 Regression analyses focusing on movement-evoked pain parameters with QuickDASH.**

|  | Crude | | Adjusted | |
|---|---|---|---|---|
|  | **B (95% CI)** | ***p*** | **B (95% CI)** | ***p*** |
| Pain VAS during daily activities | .169 (−.093 to .430) | .196 | .319 (−.011 to .648) | .057 |
| SMP | .368 (.013 to .724) | .043[*] | −.108 (−.622 to .406) | .664 |
| TSAP score | .456 (.155 to .757) | .004[**] | .461 (.099 to .824) | .015[*] |
| Adjusted R$^2$ | | .390 | | 0.440 |

**Notes.**

Adjusted for age, sex, height, weight, duration of pain, range of motion of shoulder abduction, muscle strength of shoulder flexion.

Abbreviations: VAS, Visual analogue scale; SMP, single movement pain; TSAP, Temporal Summation of Activity-Related Pain; 95% CI, 95% confidence interval.

[*]$p < .05$.

[**]$p < .01$.

and TSAP scores ($B$ (95% CI) = 0.456 (0.155−0.757), $p = 0.004$) showed a significant impact. However, after adjustment, only the TSAP score had a significant impact ($B$ (95% CI) = 0.461 (0.099−0.824), $p = 0.015$).

## DISCUSSION

In this study, we established the TSAP as a novel method for assessing the temporal summation of pain during movement in patients with SIS. We demonstrated that the TSAP score significantly correlated with cTSP and upper extremity dysfunction. In addition, regression analyses revealed that the TSAP score had a greater impact on functional outcomes than traditional assessments of movement-evoked pain. These findings contribute to the expanding literature by indicating that movement-evoked pain is a significant construct that provides unique insights into forecasting pain-related outcomes, such as dysfunction and/or disability (*Mankovsky-Arnold et al., 2014*; *Wideman et al., 2016*; *Woznowski-Vu et al., 2019*; *Hardmeier et al., 2020*).

While both TSAP and cTSP measure the temporal summation of pain, they differ fundamentally in their evaluation contexts. cTSP was conducted using experimental mechanical stimuli, mainly evaluating the hypersensitivity of multireceptive dorsal horn neurons (*Arendt-Nielsen & Graven-Nielsen, 2003*; *Meeus & Nijs, 2007*; *Staud et al., 2014*), but did not involve functional movements in daily activities. In contrast, the TSAP captures the temporal summation of pain during repetitive movements and focuses on a functional perspective of pain that is applicable to daily upper extremity tasks. Previously, some reports mentioned sensitivity to upper limb movements using repetitive lift tasks (*Sullivan et al., 2009*; *Sullivan, Larivière & Simmonds, 2010*; *Lambin et al., 2011*); however, they did not address the neural mechanisms of pain. In this regard, TSAP possibly opens up a novel field of movement-evoked pain summation, although it would be affected by peripheral conditions and sometimes it is difficult to complete consistent rhythmic stimulation such as cTSP.

In the current study, only TSAP was nominated as a factor explaining QuickDASH, suggesting that the TSAP score provided more advantages than the SMP and cTSP evaluations in terms of functional impairment. The TSAP score reflects pain intensity through actual repetitive movements and accounts for cumulative pain responses. Pain processing during movement is quite complex, involving both peripheral and central mechanisms (*Diercks et al., 2014*), and the TSAP has a more significant impact on daily activities than the SMP. This approach is partly consistent with a previous study indicating that task-specific pain assessments provide a more accurate reflection of functional limitations (*Wideman et al., 2016*).

Clinically, TSAP can help patients suffering from movement-evoked pain. When identifying patients with heightened TSAP scores, clinicians can provide tailored approaches, including education, exercise, pharmacological treatment, and surgery, to manage pain more effectively. In addition, it will be useful to monitor the response to such treatments, since the TSAP score demonstrated more linear regression with functional status than the SMP and cTSP. Further studies should investigate the relationship among TSAP scores, psychosocial factors, and treatment responses to provide more promising insights for managing patients with musculoskeletal pain.

This study has several limitations. First, this was a cross-sectional study, which prevented us from referring to causal relationships between movement-evoked pain assessments and upper-extremity dysfunction. Longitudinal studies are required to establish these causal relationships. Secondly, our sample included patients with various SIS-related diagnoses. This heterogeneity may have increased the variability in our results. Third, we used a limited number of questionnaires to evaluate upper extremity function and psychosocial problems. Therefore, a more comprehensive approach is required in future studies.

## CONCLUSIONS

The TSAP provides valuable insights into the functional impact of pain by capturing pain responses during repetitive upper limb tasks in patients with SIS. Our findings suggest that TSAP may offer a more sensitive evaluation of movement-evoked pain compared to conventional assessments, although further validation is needed.

## ACKNOWLEDGEMENTS

The authors utilized AI for initial brainstorming and the preparation of preliminary drafts. All data interpretation and final manuscript revisions were conducted by human researchers. We thank Yutaro Hyodo, Yuta Murata, and Takuma Hori for assistance with data collection.

### Funding
This work was supported by JSPS KAKENHI Grant Number JP24K20453. The funders had no role in study design, data collection and analysis, decision to publish, or preparation of the manuscript.

### Grant Disclosures
The following grant information was disclosed by the authors:
JSPS KAKENHI: JP24K20453.

### Competing Interests
The authors declare there are no competing interests.

### Author Contributions
- Katsuyoshi Tanaka conceived and designed the experiments, performed the experiments, analyzed the data, prepared figures and/or tables, authored or reviewed drafts of the article, and approved the final draft.
- Shota Oda conceived and designed the experiments, performed the experiments, authored or reviewed drafts of the article, and approved the final draft.
- Masashi Izumi conceived and designed the experiments, authored or reviewed drafts of the article, and approved the final draft.

### Human Ethics
The following information was supplied relating to ethical approvals (i.e., approving body and any reference numbers):

The Human Research Ethics Review Committee of Bukkyo University granted Ethical approval to carry out the study (Ethical Application Ref: 2023-18-A)

### Data Availability
Raw data is available as a Supplementary File.

### Supplemental Information
Supplemental information for this article can be found online at http://dx.doi.org/10.7717/peerj.19638#supplemental-information.

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
