# Peer review of "A novel evaluation approach for functional impairment in subacromial impingement syndrome: focus on Temporal Summation of Activity-related Pain (TSAP)"

_PeerJ, doi:10.7717/peerj.19638_

## Round 0.1 · original submission · Minor Revisions

Reviewer 1 ·

Basic reporting

No comment

Experimental design

The ethical approval and the informed consent taken from the participants should be mentioned prior to the recruitment of the participants in the methods section. Please specify who will be diagnosing the cases of Subacromial Impingement Syndrome. The supporting references from where the idea was obtained for measuring Single-movement-pain and TSAP needs to be mentioned. Overall, the article is very useful in bridging the knowledge gap, and has good replicability.

Validity of the findings

No comment

Additional comments

The article is a novel approach to assess TSAP, and would be very much useful in assessing as well as managing SIS.

·

Basic reporting

The manuscript entitled "A novel evaluation approach for functional impairment in subacromial impingement syndrome: focus on temporal summation of activity-related pain (TSAP)" has been reviewed. This cross-sectional study addresses an interesting and clinically relevant topic. The manuscript is written in clear and professional English. The introduction provides a solid background on subacromial impingement syndrome and the limitations of conventional pain assessments. The rationale for developing the TSAP is well-articulated. The manuscript is well-referenced, incorporating relevant prior studies to support its claims.

Experimental design

The research question is meaningful and fills an existing knowledge gap. The methodology is described in sufficient detail. However, the diagnostic criteria for shoulder impingement syndrome requires further clarification. "(2) a positive result on at least one of the orthopedic tests performed by a physical therapist, including the Hawkins impingement sign, NeerĂ­s impingement sign, painful arc sign, and/or Empty can test [10]" is inconsistent with what the referenced article suggested. Please carefully review the results and conclusions of the reference (https://doi.org/10.1016/j.apmr.2009.05.015), and justify the diagnostic criteria used in your study.

Validity of the findings

The data is robust, and statistical analyses are appropriate. The authors claim TSAP is "superior" to conventional movement-evoked pain assessments, which appears to be overstated. The claim should be tempered unless a direct comparative validation is performed.

Reviewer 3 ·

Basic reporting

Though the article is well written, hypothesis should be clearly mentioned in the text.

Experimental design

There are a few observations regarding the exclusion criteria for the study:
1. There is no mention of the other conditions of upper extremity that could be a source of pain. Authors
should clarify if this point was addressed or not.

Validity of the findings

No comment

Additional comments

Though the manuscript is well written, authors should emphasize the need of this novel approach in a separate paragraph.

---

## Round 0.2 · accepted · Accept

Dear Authors

Thank you for this important work in the area of Subacromial Impingement Syndrome. In agreement with the referees, I have decided to accept your paper

·

Basic reporting

The manuscript has been substantially improved. The authors replied satisfactorily to reviewers’ comments and took into account the suggestions given.

Experimental design

no concerns

Validity of the findings

no concerns

Reviewer 3 ·

Basic reporting

The manuscript is well written. There is a clear and professional use of English throughout the text. The introduction provides a good insight into subacromial impingement syndrome. Authors have added the hypothesis as suggested.

Experimental design

Methods have been explained in detail. Study methods have been explained in detail.

Validity of the findings

No comment